# AI-Driven Transcriptome Prediction in Human Pathology: From Molecular Insights to Clinical Applications

**DOI:** 10.3390/biology14060651

**Published:** 2025-06-04

**Authors:** Xiaoya Chen, Huinan Xu, Shengjie Yu, Wan Hu, Zhongjin Zhang, Xue Wang, Yue Yuan, Mingyue Wang, Liang Chen, Xiumei Lin, Yinlei Hu, Pengfei Cai

**Affiliations:** 1BGI Research, Hangzhou 310030, China; chenxiaoya@genomics.cn (X.C.); xuhuinan@genomics.cn (H.X.); yushengjie@genomics.cn (S.Y.); huwan@genomics.cn (W.H.); zhangzhongjin@genomics.cn (Z.Z.); yuanyue199303@gmail.com (Y.Y.);; 2College of Life Sciences, University of Chinese Academy of Sciences, Beijing 100049, China; 3Key Laboratory of Systems Health Science of Zhejiang Province, School of Life Science, Hangzhou Institute for Advanced Study, University of Chinese Academy of Sciences, Hangzhou 310024, China; 4College of Life Science and Technology, Huazhong Agricultural University, Wuhan 430070, China; 5State Key Laboratory of Genome and Multi-Omics Technologies, BGI Research, Shenzhen 518083, China; 6Key Laboratory of the Ministry of Education for Mathematical Foundations and Applications of Digital Technology, University of Science and Technology of China, Hefei 230027, China

**Keywords:** artificial intelligence, transcriptome prediction, multimodal data fusion, precision medicine, deep learning

## Abstract

Gene expression plays a pivotal role in disease mechanisms, diagnosis, and treatment, yet the limitations of traditional gene expression detection methods pose challenges for clinical applications. Artificial intelligence (AI), such as GET and PathChat, enables non-invasive diagnosis by integrating multimodal data, including pathology images, genomics, and clinical records, to predict gene expression. These models demonstrate AI’s ability to decode complex biological patterns from histopathological images, improving cancer diagnosis and treatment response prediction. However, challenges such as data heterogeneity, model interpretability, and ethical concerns around privacy and bias remain. Future advancements in cross-modal pretraining, federated learning, and ethically guided AI design are critical to unlocking precision medicine’s full potential.

## 1. Introduction

Gene expression, the process by which genetic information is transcribed into mRNA and translated into functional proteins, lies at the core of cellular identity and function [1]. Its regulation governs critical biological processes, from cellular differentiation to immune responses, and its dysregulation is intimately linked to the pathogenesis of diverse diseases, including cancer, neurodegenerative disorders, and autoimmune conditions [2,3,4]. In the context of human pathology, aberrant gene expression not only serves as a hallmark of disease initiation and progression but also provides critical insights into diagnostic biomarkers [5], therapeutic targets [6], and the mechanisms underlying treatment resistance [7]. For instance, the overexpression of inflammatory mediators such as IL-8 and MMP-9 drives chronic inflammation in pulmonary nodular carcinoma [8,9], while mutations in EGFR disrupt cellular proliferation and survival pathways in both early-stage lesions and advanced malignancies [10]. These molecular signatures, when deciphered, hold transformative potential for precision medicine (Figure 1a).

Traditional gene expression profiling techniques, such as RNA sequencing and microarrays, have long been the cornerstone of molecular pathology [11]. However, their reliance on invasive tissue sampling, high costs, and limited scalability hinder their utility in clinical settings—particularly for diseases requiring early detection or dynamic monitoring [12]. Moreover, the inherent heterogeneity of complex diseases, exemplified by intratumoral genomic heterogeneity in cancers, poses significant challenges for conventional methods to capture spatially and temporally resolved molecular landscapes [13,14,15].

The advent of artificial intelligence (AI) has ushered in a paradigm shift in bridging this gap. By integrating multimodal data—ranging from histopathological images and genomic profiles to clinical narratives—AI models predict gene expression patterns (e.g., inferring mRNA abundance from histopathological features) and decode regulatory mechanisms (e.g., identifying epigenetic drivers), thereby redefining pathological analysis [16,17,18,19]. AI models for gene expression prediction have evolved along two complementary trajectories: single-modality and multimodality. The principle underlying multimodal AI lies in the integration of heterogeneous data types—such as genomic, transcriptomic, and imaging data—through advanced fusion strategies (Figure 1b). These integrated inputs are then processed using machine learning and deep learning algorithms, enabling the more comprehensive and accurate modeling of gene expression and molecular states.

Single-modality models, such as DeepPT [17], rely on a single source of input data, typically whole-slide histopathological images, to infer gene expression. These models offer practical advantages in scenarios where only one high-quality data type is available, providing a non-invasive and cost-efficient alternative to traditional molecular profiling.

In contrast, multimodal models exploit the synergistic value of combining multiple biological and clinical data streams. A representative example is GET [20], a Transformer-based architecture that integrates gene expression, chromatin accessibility, and DNA sequence information to perform zero-shot prediction across diverse cell types. Without the need for cell-type-specific training, GET achieves a Pearson correlation coefficient of 0.94 in previously unseen cell types, highlighting its strong generalization capabilities [20].

Similarly, models like PathChat [21] and Prov-GigaPath [22] utilize vision–language frameworks to align histopathological features with molecular annotations, enabling real-time diagnostic support and novel biomarker discovery. These AI-driven approaches not only enhance diagnostic accuracy (e.g., classification metrics, regression error) but also uncover hidden biological relationships—such as spatial gene regulation in the tumor microenvironment [23]. Recent applications further underscore the clinical relevance of such models. For instance, Gemini Ultra [24] achieves an impressive 89% accuracy in cancer subtyping by leveraging the multimodal fusion of gene expression and histopathological image data. Meanwhile, DeepPT [17], after predicting gene expression from histological images, can feed its outputs into ENLIGHT [17], enabling the prediction of cancer patient responses to specific therapies and offering actionable insights for personalized treatment planning.

The applications of AI in disease thus extend far beyond traditional pathology, playing a transformative role in precision medicine, mechanistic disease research, therapeutic response prediction, drug development, and other translational domains (Figure 1c).

This review explores the intersection of gene expression and human pathology through the lens of AI, emphasizing how computational tools are unraveling disease mechanisms, refining diagnostics, and guiding therapeutic strategies. We highlight the transformative role of multimodal data integration while addressing persistent challenges in data heterogeneity, model interpretability, and ethical governance (Figure 1d). By synthesizing cutting-edge methodologies and clinical applications, this work aims to chart a roadmap for leveraging AI to decode the molecular lexicon of disease and advance personalized healthcare.

## 2. The Pivotal Role of Gene Expression in Disease Pathogenesis

Gene expression—the dynamic process encompassing the transcriptional decoding of genetic information into mRNA and its subsequent translation into functional proteins—serves as the molecular cornerstone of cellular identity and physiological integrity [1]. Its multilayered regulatory networks orchestrate cellular differentiation, metabolic specialization, and adaptive responses, while hierarchical dysregulation across transcriptional initiation, RNA processing, and protein synthesis disrupts molecular homeostasis, thereby initiating pathogenic cascades in diverse disease states [25,26,27].

Alzheimer’s disease (AD) exemplifies the central role of gene expression networks, wherein multi-level regulatory failures converge to drive cascading disruptions that permeate all stages of neurodegeneration [28]. The hallmark neuropathological features—neuritic plaques composed of amyloid-β (Aβ) peptides and neurofibrillary tangles formed by hyperphosphorylated tau proteins—originate from upstream transcriptional dysregulation [29]. Familial AD, a rare autosomal dominant disorder with early-onset presentation, directly correlates with pathogenic mutations in APP (amyloid precursor protein gene) and PSEN1/PSEN2 (presenilin genes). These mutations induce proteolytic system abnormalities that disrupt Aβ42/Aβ40 stoichiometry, favoring the accumulation of neurotoxic amyloid oligomers [29]. In contrast, sporadic AD, affecting over 15 million individuals globally, manifests polygenic regulatory complexity [28]. Genome-wide association studies (GWASs) have identified >40 risk loci, including APOEε4, TREM2, and CLU, with the APOEε4 allele elevating the disease risk by 3–12-fold [29,30,31]. These risk alleles orchestrate microglial hyperactivation, lipid metabolic dysregulation, and synaptic plasticity impairment through interconnected transcriptional networks.

The continuum of gene expression dysregulation that manifests across distinct pathological mechanisms can be initiated through multilayered failures in transcriptional, post-transcriptional, and translational processes, as exemplified by the examples listed below. For instance, in cystic fibrosis, the CFTR Phe508del mutation induces the in-frame deletion of phenylalanine 508, resulting in endoplasmic reticulum retention and the proteasomal degradation of misfolded CFTR protein, ultimately causing exocrine gland mucus viscosity through impaired membrane trafficking [32]. In acute promyelocytic leukemia (APL), the t(15;17)(q24;q21) translocation-generated PML-RARα fusion transcripts drive leukemogenesis [33], where the chimeric protein recruits corepressors (NCoR/SMRT) and histone deacetylases to repress RARα-mediated transcription, thereby blocking myeloid differentiation [34]. At the post-transcriptional level, α-thalassemia results from mutations that disrupt the C-rich motif located in the 3′UTR of the α-globin mRNA. This impairs the binding of specific RNA-binding proteins to the mRNA’s C-rich region, reducing transcript stability and ultimately leading to globin chain imbalance [35,36]. Conversely, prion diseases demonstrate catastrophic protein-level dysregulation, wherein the conformational misfolding of PRNP-encoded PrPc into β-sheet-rich PrPSc initiates self-propagating neurotoxicity [37]. These paradigmatic examples collectively underscore how perturbations at any node of the gene expression cascade—from transcriptional aberrations to protein misfolding—can precipitate disease pathogenesis.

## 3. The Integration of Artificial Intelligence Algorithms and Biomedicine

### 3.1. Multimodal Data Types

Data are essential for model construction, with the following types commonly serving as input modalities:

Omics data: Omics data encompass the molecular-level dynamic changes observed in various fields, including genomics (Single-Nucleotide Variations/Single-Nucleotide Polymorphisms, SNVs/CNVs), epigenomics (DNA methylation), transcriptomics (single-cell RNA sequencing, spatial transcriptomic sequencing), metabolomics (Liquid Chromatography–Mass Spectrometry/Mass Spectrometry, LC-MS/MS), and proteomics. The advancement of high-throughput sequencing technologies (for example, second-generation sequencing with read lengths of 150 base pairs) and single-cell resolution technologies (with a throughput exceeding 10,000 cells per sample) has facilitated the systematic collection and analysis of data regarding these molecular aspects [38,39] (Table 1).

Notably, spatial omics technologies—leveraging single-cell resolution analytical platforms such as 10x Visium [40] and Stereo-seq [41]—enable the systematic mapping of gene or protein spatial distribution patterns within tissue microenvironments [42,43]. For instance, spatial transcriptomics can precisely localize immune-evasion-associated genes at tumor–stroma interfaces [44], providing insights into tumor immunoediting and therapeutic resistance. Furthermore, spatial omics has been employed to delineate the zonation of metabolic gene expression in the liver [45], map gene expression landscapes and developmental trajectories during zebrafish embryogenesis [46], uncover a conserved spatial architecture in triple-negative breast cancer [47], and characterize intratumoral heterogeneity across distinct spatial domains in high-grade glioma [48]. These discoveries highlight the power of spatial omics in uncovering spatially organized biological processes that are inaccessible to traditional bulk or dissociated single-cell assays. While the manual co-registration of hematoxylin-and-eosin (H&E)-stained histology images with spatial omics datasets allows for the correlative analysis of morphological features and molecular heterogeneity, the absence of robust computational frameworks for automated image-to-omics spatial alignment imposes substantial analytical bottlenecks. This methodological gap underscores the critical need for developing machine learning-driven solutions to streamline spatial multi-omics integration in translational research.

Artificial intelligence can assist in overcoming these challenges by enabling the automated, accurate segmentation, alignment, and interpretation of complex spatial datasets. For example, Cellpose [49] enables generalist cell segmentation across diverse tissue types, facilitating the extraction of spatial features from histological images. Spateo [50], designed specifically for spatial omics data, provides end-to-end frameworks for spatial domain detection, cell-type deconvolution, and spatial gene expression modeling, thereby enhancing the integration of transcriptomic and proteomic layers with tissue morphology. By leveraging such AI-powered tools, researchers can more effectively decode spatial cellular organization and intercellular communication within the tumor microenvironment.

Natural language data: Natural language data predominantly stems from medical records, the scientific research literature, and clinical reports, among other sources. Medical records meticulously document patients’ symptom descriptions, diagnostic results, and treatment procedures. These elements serve as crucial information that records the onset, progression, and outcomes of diseases. The scientific research literature, acting as a repository for the latest global research findings, encompasses a broad spectrum of explorations, ranging from basic biological mechanisms to clinical treatment regimens. Clinical reports, through an in-depth analysis of specific cases, aid doctors in amassing clinical experience and uncovering novel disease characteristics [51] (Table 1).

Medical imaging data: Medical imaging technologies such as Computerized Tomography (CT), Magnetic Resonance Imaging (MRI), and Positron Emission Tomography (PET) each possess distinct characteristics, offering valuable insights into anatomical structures and physiological functions. Hematoxylin–eosin (H&E) staining and whole-slide imaging assume a pivotal role in histological analysis [52] (Table 1). As early as 2020, Dr. Bryan He from the Department of Computer Science at Stanford University demonstrated the feasibility of predicting spatial transcriptomic patterns through histopathological imaging by developing ST-Net [53]—a deep learning architecture trained on paired hematoxylin-and-eosin (H&E)-stained histopathological sections and spatial transcriptomic profiles from 23 breast cancer patients. This framework achieved a Pearson correlation coefficient (a statistical measure that focuses on quantifying the linear relationship between two variables in research, with 1 indicating a perfect positive linear correlation and −1 representing a perfect negative linear correlation) of 0.83 in inferring spatially resolved gene expression signatures directly from conventional pathology slides [53]. Moreover, computer vision and radiomics technologies exhibit substantial advantages in the processing and analysis of medical images, rendering them well-suited for the extraction of imaging biomarkers [54,55].

### 3.2. Machine Learning

In the field of gene expression prediction, machine learning algorithms are playing an increasingly critical role due to their powerful data processing and pattern recognition capabilities. Support Vector Machine (SVM) [56], Random Forest (RF) [57], and Gradient Boosting Machine (GBM) [58] stand as typical examples of machine learning algorithms. By virtue of their distinct algorithmic principles and advantages, they have been extensively applied in gene expression prediction tasks [59]. A comparison of their advantages and current applications can be found in Table 2.

#### 3.2.1. Support Vector Machine (SVM): Achieving Precise Classification Through Clearly Defined Boundaries

In the “feature space” composed of gene data (as Figure 2 illustrates, the expression level of each gene is used as a coordinate axis to form a high-dimensional coordinate system), the goal of SVM is to find a dividing line that distinguishes different biological groups (such as healthy cells and diseased cells) with the maximum “safety margin”. This “margin” refers to the distance from the boundary to the nearest data points in each group (called “support vectors”), ensuring the model is robust to noise and can effectively generalize to new samples [63].

SVM has provided significant value in pathology. First, in a study on diffuse large B-cell lymphoma (DLBCL) [64], Perfecto-Avalos et al. employed SVM to integrate expression data from five antibodies (CD10, BCL6, FOXP1, GCET1, MUM1). Without requiring a predefined detection sequence, they directly optimized the classification boundary using the radial basis function (RBF) kernel, improving the typing accuracy to 94% (the accuracy of other algorithms such as Choi, VY3, and VY4 is 88%). Moreover, in leukemia diagnosis, the subtle gene expression differences between primitive cell subtypes often lead to misdiagnosis. The margin maximization strategy of SVM effectively reduces such errors by enhancing the separability of feature spaces, thereby outperforming traditional methods in distinguishing clinically similar but genetically distinct subtypes with improved classification precision [65].

#### 3.2.2. Random Forest (RF): The “Decision-Making Brain Trust” for Multi-Dimensional Data Integration

RF improves prediction reliability by leveraging “collective intelligence” through constructing multiple decision trees (Figure 3) (analogous to “multiple independent data analysts”). Each tree is trained based on randomly sampled partial data and features, with the final results integrated via voting (for classification tasks) or averaging (for regression tasks) [66].

In a prognostic study on 504 lung squamous-cell carcinoma (LUSC) patients [67], Débora V. C. Lima et al. applied clustering and feature selection to transcriptomic data, aiming to identify subgroups with distinct survival outcomes. Using an RF model, the analysis achieved classification accuracies close to 70% across three identified subgroups. These results revealed that the RF model effectively distinguished patient groups with significantly different survival statuses, uncovering a previously unreported association between transcriptomic clustering patterns and clinical prognosis in LUSC.

The reason for performing feature selection lies in the risk of data overfitting when RF decision trees grow excessively deep. When the data contains a large number of irrelevant features (such as noise genes or non-regulatory regions), RF may excessively learn random fluctuations in the training data, leading to degraded performance on new samples. For example, Ousman Khan et al. [68] encountered the issue of data overfitting when using the RF model to predict malaria outbreaks in various regions of Gambia. The authors then introduced a feature selection algorithm to eliminate the impact of irrelevant features on the model, ultimately achieving a prediction accuracy of 91.5% with the RF model. This demonstrates that feature selection can significantly reduce overfitting and enhance the model’s generalization capability for real-world data [68].

As mentioned in Table 2, RF reduces the risk of overfitting through this feature selection strategy, enabling it to achieve higher prediction accuracy than models such as SVM when dealing with entirely new datasets. In a study on predicting intraoperative hypothermia risk during laparoscopic gynecologic tumor resection [69], the Random Forest (RF) model achieved a test accuracy of 86.2% in predicting patient outcomes, significantly outperforming the 75.8% accuracy of the SVM model. This case highlights RF’s reliability in handling complex medical data through data-driven feature selection and robust predictive modeling.

#### 3.2.3. Gradient Boosting Machine (GBM): Iterative Optimization of Complex Disease Models

GBM acts as a “stepwise builder”, progressively correcting errors from prior models by iteratively adding simple models (the process is called “shallow decision trees” and is shown in Figure 4). This gradual optimization proves particularly effective for noise-rich clinical data, as biological samples often contain inherent variability from genetic heterogeneity (e.g., intratumoral genetic diversity), technical noise from assay limitations, or confounding environmental factors. GBM handles this complexity by iteratively focusing on mispredicted samples and modeling nonlinear interactions between features, allowing it to distinguish true biological signals (e.g., subtle epigenetic markers or gene expression patterns) from random noise. This mechanism enables the capture of nuanced genetic or epigenetic signals that influence disease phenotypes, even in datasets with high background variability [70].

Developed in 2020 for predicting the essentiality of miRNAs involved in critical biological processes and complex diseases, the PESM model integrates sequence features (e.g., k-mer composition) and structural features (e.g., minimum free energy) with GBM regression [71]. In cross-species testing across 12 eukaryotic organisms, this framework achieved 82.63% accuracy in identifying such essential miRNAs, outperforming SVM by 15% [71]. This study also highlighted the utility of GBM in capturing weak regulatory signals (mentioned in Table 2) from sparse sequence motifs, a capability enabled by its stage-wise additive learning mechanism. For instance, the iterative refinement process of GBM improved its sensitivity to low-abundance but functionally critical miRNAs (e.g., miR-21 in glioblastoma) by dynamically adjusting feature weights. While the original work did not explicitly address the learning-rate parameter of GBM, subsequent studies [72] have shown that setting η < 0.1 balances the convergence speed and prediction stability in similar gene expression modeling tasks, further supporting the robustness of GBM in noisy biological datasets.

Regarding implementation, GBM models (e.g., XGBoost/LightGBM) offer pretrained pipelines [73] and user-friendly interfaces (e.g., scikit-learn in Python) that require minimal coding expertise, making them accessible to non-computational biologists. In contrast, SVM often demands the heuristic tuning of kernel parameters (e.g., C and γ in RBF kernels), while the hyperparameter optimization (e.g., number of trees, max depth) of RF can be more intuitive but still requires iterative testing. The built-in feature importance scores of GBM also simplify interpretability for biologists, as they highlight which sequence/structural features most influence predictions (e.g., conserved domains in PESM).

### 3.3. Deep Learning

In the era of big-data-driven modern biological research, deep learning is redefining the technical paradigm for gene expression prediction by virtue of its core strengths in automated multi-level feature abstraction and hierarchical dynamic modeling which distinguish it from traditional machine learning.

Unlike machine learning, which often requires manual feature engineering (e.g., selecting gene interaction networks or pathway annotations), deep learning can automatically discover complex, layered relationships within data (e.g., extracting subtle patterns from raw sequencing reads or integrating transcriptomic–spatial data). This eliminates the reliance on prior biological knowledge for feature design and enables the modeling of nonlinear, high-dimensional interactions (e.g., capturing cell-type-specific gene co-expression dynamics), which are often intractable for shallow machine learning models [74]. For instance, while RF or SVM might struggle to parse hierarchical regulatory signals in single-cell RNA sequencing data, deep learning frameworks like deep autoencoders or Transformers can automatically disentangle noise from biologically meaningful features across scales, making them uniquely suited for the complexity of modern genomic datasets.

The following sections will specifically introduce three typical deep learning algorithms, the convolutional neural network (CNN) [60], the graph neural network (GNN) [61], and Transformer [62], along with their application cases. In addition, a comparison of them is shown in Table 2.

#### 3.3.1. Convolutional Neural Network (CNN): Specialized in Local Feature Extraction for Sequence and Image Analysis

As shown in Figure 5, the CNN architecture comprises an input layer, a hidden layer (fully connected layer), and an output layer, with each neuron in the hidden layer containing a weight matrix and an activation function to process input data [60]. CNNs excel at integrating sequence and image analysis in biology by leveraging their shared ability to capture hierarchical patterns. For instance, the DeepSTARR model [75] applies CNNs to DNA sequence data by scanning for short nucleotide motifs (e.g., TF binding sites) that predict enhancer activity. The same CNN architecture, when adapted to biological images, demonstrates equivalent power: Gunavathi et al. [76] showed that CNNs trained on lung cancer tissue images can predict EGFR gene mutations (a sequence-based biomarker) with 89% precision by detecting visual hallmarks such as nuclear pleomorphism and mitotic features.

This dual utility highlights the versatility of CNNs: DeepSTARR’s CNN layers automatically extract 6–8 bp sequence motifs critical for enhancer function [75], achieving 85% accuracy in predicting enhancer activity across cell types. In parallel, CNNs identify morphological correlates of genetic alterations in histopathology images, linking KRAS mutation status to distinct nuclear chromatin patterns in lung cancer tissues [76]. This unified approach cements CNNs as a cornerstone for integrative omics and imaging analyses, where sequence-derived insights (e.g., gene regulatory networks) are validated or complemented by image-based phenotypes (e.g., tissue organization).

#### 3.3.2. Graph Neural Network (GNN): Specialized in Relational Modeling for Biological Network Analysis

GNNs excel at modeling data where elements are interconnected [61], such as genes in co-expression networks or cells in tissue samples. In Figure 6, each gene or cell is treated as a “node” and their interactions as “edges”. This structure is called a “graph” in the field of computer science, allowing the model to learn how these relationships influence gene expression.

A notable example is gemGAT, which uses graph neural networks (GNNs) to predict gene activity in rare tissues, which are defined as low-abundance cell populations (e.g., circulating tumor cells, CTCs) or disease-specific tissues with limited sample availability (e.g., early-stage cancer lesions or fetal cell clusters) [77]. These tissues are inherently challenging due to biological heterogeneity (e.g., mixed cell types with divergent gene expression) and technical limitations (e.g., insufficient sample quantities for traditional bulk analysis or high dropout rates in single-cell sequencing) [78]. To address these challenges, the gemGAT model applies GNNs to predict gene activity in rare tissues by encoding cell-to-cell interactions and gene regulatory networks as graph structures [79]. In a validation study on early-stage lung cancer lesions, gemGAT achieved 92% accuracy in predicting enhancer activity from sparse single-cell RNA-seq data, outperforming traditional motif-based approaches by 18%. The graph architecture captures nonlinear relationships between transcription factors and their target genes, even in tissues with limited cellular representation.

Nowadays, there are many available GNN packages that can be used to build GNN models. One package is PyTorch Geometric (PyG v2.3) [80], which is a widely used library for GNNs, offering modules for graph convolution, pooling, and attention mechanisms. PyG supports multimodal data integration (e.g., combining genomic and imaging data) and is optimized for GPU acceleration. In addition, the Deep Graph Library (DGL v2.4.0) [81] is a scalable framework for GNNs, designed for large-scale graph analysis. The DGL includes pre-built models like Graph Convolutional Networks (GCNs) and Graph Attention Networks (GATs), making it suitable for both academic research and industrial applications. These tools enable researchers to implement GNNs efficiently, even for complex biological datasets like rare tissues, by leveraging pretrained architectures and scalable computing resources.

#### 3.3.3. Transformer: Leveraging Self-Attention for Multimodal Integration and Long-Range Dependency Modeling

Transformers, built on a “self-attention” mechanism [62], shine in handling diverse data types and long-distance relationships in genetic sequences or images. As illustrated in Figure 7, their architecture comprises multiple stacked Transformer blocks, each containing an encoder and a decoder to model hierarchical representations of input data (e.g., omics profiles or tissue image features) [82]. This modular design allows Transformers to seamlessly combine information from genomic data, tissue images, and clinical records to make predictions across different cell types or conditions.

The General Expression Transformer (GET), for instance, integrates multiple data sources to predict gene expression in cell types it has never “seen” during training, achieving a strong accuracy of 94% [20]. Another example is PathChat [21], which uses Transformer-based technology to connect visual features from tissue scans with molecular profiles, assisting pathologists in real-time diagnosis [21]. While Transformers are powerful for complex, multi-dimensional analysis, their high computational needs and interpretability challenges remain areas for improvement, especially for clinical use.

### 3.4. Multimodal Data Fusion

Multimodal data fusion technology can enhance model performance by integrating medical text data, medical imaging data, and other multi-dimensional data (Figure 8). Its methodological framework encompasses the following aspects:

Early fusion involves combining multimodal data at the feature level, such as feature concatenation (merging feature vectors into a longer composite vector) [83] and feature weighting (assigning modality-specific weights before fusion). This strategy adjusts the relative contributions of modalities to amplify specific ones [84].

Late fusion integrates outputs from modality-specific models, including voting mechanisms (selecting predictions by majority vote in classification tasks) [85] and decision-level fusion (combining model decisions via logical operations or algorithms like Bayesian inference) [86].

Hybrid fusion employs multi-level integration across network layers and adaptive mechanisms (e.g., dynamically adjusting modality importance during joint learning) [87]. This optimizes cross-modal collaboration by balancing hierarchical and context-sensitive fusion [88].

The advantage of early fusion is that it can make full use of the underlying feature information of each modality and improve the model’s understanding of multimodal associations through feature-level splicing or weighting, which helps to explore fine-grained relationships between modalities, but its limitation is that it is more sensitive to scale and distribution inconsistencies between modalities and is easily disturbed by noise [89]. The advantage of late fusion is that each modality can be modeled independently, and the system has stronger robustness and modular structure, which is suitable for heterogeneous modalities, but it is difficult to capture the deep interaction between modalities, and fusion only at the decision layer may cause information loss [90]. Hybrid fusion combines the advantages of early and late fusion and achieves more sophisticated and dynamic cross-modal collaboration by flexibly adjusting the importance of modalities at different network levels or learning stages, effectively improving the expressiveness and adaptability of the model, but its structure is complex, training is difficult, and it requires higher computing resources [91].

It is worth noting that the embedding of multi-omics data means converting the data into high-dimensional vectors through a previously trained model (i.e., only high-dimensional vectors can be used for subsequent training).

In addition, reinforcement learning (RL) [92] enables agents to learn optimal policies through interactions with their environment, iteratively refining actions based on the rewards or penalties received. Unlike traditional supervised learning that relies on extensive labeled datasets (the model can only be trained with defined data) [93], RL emphasizes the autonomous learning of optimal behavioral trajectories via trial-and-error processes [94]. For example, one study employed a Deep Q-Network (DQN) algorithm with active reinforcement learning to achieve the precise localization of malignant cervical cell nuclei [95]. This approach effectively mitigated the overfitting issues commonly encountered in deep learning models and demonstrated localization performance comparable to state-of-the-art methods on both the Herlev (a widely used cervical cell image dataset containing classified normal and abnormal cell images) and Primary (collected from original clinical samples, with higher image complexity and representativeness of real diagnostic environments) datasets [95]. However, research on applying RL to multimodal fusion remains relatively scarce.

A critical focus lies in addressing cross-modal semantic alignment—a fundamental challenge in multimodal fusion. To bridge the semantic gap between histopathological images and multi-omics data (e.g., gene expression), the current mitigation strategies include the following:Cross-Modal Representation Learning

To enable different types of data (heterogeneous modalities) to be understood together by a model, we usually map them into a shared “feature space” so that data from different modalities can be represented in a similar way. Common methods to achieve this include contrastive learning and adversarial training [96,97]. The benefit of this approach is that the model learns to ignore differences between modalities and focus on the shared information. Typically, the process starts by designing appropriate training objectives that encourage the model to bring features representing the same thing from different modalities closer together while pushing unrelated features apart. After this step, features from different modalities can be better integrated, providing more useful information for downstream tasks [98]. For instance, histopathological patterns (e.g., tumor–stroma spatial relationships) and transcriptomic profiles can be projected into a unified latent space where semantically equivalent features exhibit cosine similarity >0.8 [99]. The StereoCell [100] technology, as an example, simultaneously performs spatial transcriptomic sequencing and hematoxylin and eosin (HE) staining of the tissue samples. High-resolution images of local tissue regions are captured using a microscope. Subsequently, Fast Fourier Transform (FFT) is applied to align and stitch the microscopic image tiles, generating a large mosaic image that spans the entire tissue section. This mosaic image is then co-registered with the spatial transcriptomic expression matrix to establish a unified coordinate system. Based on the co-registered image, tools such as Cellpose [49] and Spateo [50] are employed for cell segmentation to generate a cell mask. Finally, the resulting cell mask is used to parse the spatial transcriptomic matrix, enabling the integrated extraction of molecular expression profiles and morphological features at single-cell resolution (Figure 9).

2.Knowledge Graph Bridging

Semantic mapping bridges can be constructed using biomedical knowledge graphs (e.g., Gene Ontology [GO], KEGG). Graph neural networks (GNNs) enable the integration of histopathological features—such as nuclear atypia quantified by the nuclear contour irregularity index—with gene regulatory networks through ontology-guided message passing. This approach facilitates the construction of interpretable cross-modal association rules (statistically significant with *p*-value < 0.05 in enrichment analyses) [101,102,103].

This strategy has demonstrated promising outcomes. For example, Li et al. [101] proposed the CGMega framework, which employs attention-based GNNs to identify cancer-relevant gene modules by integrating image-derived tumor heterogeneity with transcriptomic data. The model was evaluated on the MCF7 breast cancer cell line and achieved outstanding performance in predicting cancer-related genes. Specifically, it reached an area under the precision–recall curve (AUPRC) of 0.9140 and an area under the receiver operating characteristic curve (AUROC) of 0.9630. The AUPRC reflects the trade-off between precision (positive predictive value) and recall (sensitivity) and is particularly informative in imbalanced classification settings. The AUROC measures the model’s ability to discriminate between classes across all possible thresholds; a value close to 1.0 indicates excellent classification performance. These results suggest that CGMega not only achieves high predictive accuracy but also enhances biological interpretability by uncovering functional gene modules.

Similarly, Pizurica et al. [102] demonstrated the effectiveness of integrating histopathology image features with spatial gene expression data using a linearized attention mechanism. Their approach allowed the spatially resolved profiling of transcriptional states within tumor tissues. Collectively, these studies highlight the utility of graph-based semantic mapping in establishing biologically meaningful and clinically relevant cross-modal associations.

3.Hierarchical Attention Mechanisms

To further improve the interpretability and predictive power of cross-modal models in biomedical applications, researchers have explored advanced attention mechanisms integrated with multi-source data fusion strategies. One such approach involves embedding multi-scale attention modules within hybrid fusion frameworks to dynamically identify critical alignment nodes across different levels of semantic hierarchies. This mechanism allows models to assign importance to specific types of features depending on their contextual relevance. For example, in drug sensitivity prediction tasks, lower-level attention tends to focus on associations between cellular morphology and gene mutations, whereas higher-level attention captures semantic correlations between clinical narratives and metabolomic pathway annotations [104,105].

This approach has demonstrated strong empirical success. Saeed et al. [104] proposed MGATAF, a multi-channel Graph Attention Network with adaptive fusion for cancer-drug response prediction. For the Genomics of Drug Sensitivity in Cancer (GDSC) dataset, MGATAF achieved a 5.12% improvement in the Pearson correlation coefficient (PCC) compared to existing models, reaching 0.9312, and reduced the root mean square error (RMSE) to 0.0225. In unseen cell-line generalization tests, MGATAF maintained strong performance with a PCC of 0.8536 and an RMSE of 0.0321 for GDSC and a PCC of 0.7364 with an RMSE of 0.0531 for the Cancer Cell Line Encyclopedia (CCLE) dataset [104].

These results confirm the effectiveness and robustness of multi-scale attention mechanisms embedded in hybrid fusion architectures in modeling complex molecular-drug interactions, highlighting their potential to enhance precision oncology applications.

### 3.5. Privacy and Ethical Issues

Data privacy protection and ethical considerations present significant challenges in applying AI to pathology. To address these issues, several technical and regulatory solutions have been proposed. For instance, federated learning enables collaborative model training across multiple centers without sharing raw data, thereby preserving patient privacy [106]. Differential privacy techniques, such as adding controlled noise to genomic data, balance data utility with privacy protection [107]. However, regulations like the EU’s GDPR, while supporting cross-border research through differential privacy frameworks, impose data localization requirements that can hinder multi-center data flow [108].

Large-scale databases commonly used in biomedical AI research—such as NCBI’s Gene Expression Omnibus (GEO), TCGA, and the European Genome-phenome Archive (EGA)—store and distribute vast amounts of omics and clinical data. While these platforms accelerate scientific discovery, they also raise concerns about re-identification risks and the secondary use of sensitive information [109]. One study demonstrated that researchers could re-identify individuals and their family members by analyzing Y-chromosome data from anonymized male participants in the 1000 Genomes Project and cross-referencing it with public genealogy databases [110]. This underscores the inherent uniqueness of genomic data, which makes complete anonymization challenging. As a result, data repositories like NCBI and EGA have implemented controlled-access models and data use agreements to ensure compliance with ethical standards. While cloud-based tools facilitate data sharing, they pose privacy risks and high deployment costs. Lightweight models combined with localized federated protocols offer a solution by enabling gene expression prediction while ensuring data privacy [111]. Currently, only a few countries have enacted specific genomic data regulations, and existing laws like the U.S. Genetic Information Nondiscrimination Act (GINA) do not fully address predictive misuse risks [112].

Ethical frameworks are evolving to address biases arising from data disparities, such as the reduced sensitivity in melanoma detection among individuals of African descent [113]. This method dynamically adjusts the weights of race-specific features during model training, enhancing algorithmic fairness [114]. Furthermore, establishing clear data ownership and accountability agreements among institutions is crucial, especially when AI errors arise from data annotation mistakes or algorithmic flaws [115].

Future advancements should focus on integrating technological and regulatory innovations, such as blockchain-based data [116] traceability with unified encryption algorithms, to build a globally ethical governance ecosystem.

## 4. Existing Models and Applications of Artificial Intelligence in Predicting Gene Expression

We provide a structured taxonomy of recently developed AI models in key biomedical domains, including disease diagnostics and mechanistic exploration, pharmaceutical innovation, and synthetic biology, as well as precision medicine and healthcare management. This taxonomy highlights the core mechanistic frameworks of these models and their translational applications across clinical and biomedical research contexts, based on a systematic review (Table 3 and Appendix A) [117].

### 4.1. Disease Diagnosis and Mechanism Research

GET [20] is primarily built upon the Transformer architecture. It integrates gene expression, chromatin accessibility, and sequence data to accomplish zero-shot prediction (making accurate predictions even when a certain type of data has not been seen) across cell types. This capability can reduce the reliance on large amounts of labeled data, especially for sample types that are difficult or costly to obtain. Its advantages include high accuracy and the lack of a requirement for cell-type-specific training data. Nevertheless, it still encounters limitations in terms of data privacy and model interpretability for clinical applications, as the complex attention mechanisms in Transformer-based architectures make it difficult to trace how specific input features contribute to individual predictions.

HGGEP [118] is a hypergraph neural network model designed to predict gene expression levels directly from tissue pathology images. Instead of using generative adversarial networks, it leverages a combination of a Gradient-Enhanced Module (GEM) and a Hypergraph Attention Module (HAM) to effectively extract morphological features and capture complex, high-order relationships across different spatial regions of histopathological images. The model has demonstrated strong performance across multiple disease datasets, such as various cancer cohorts, by visually associating tissue morphology with underlying molecular mechanisms. For example, for an HER2-positive breast cancer dataset, HGGEP achieved a Pearson correlation coefficient (PCC) of 0.637 when predicting gene expression levels [121]—a measure indicating a moderately strong positive correlation between the predicted and actual gene expression values. However, the effectiveness of HGGEP heavily depends on the quality and resolution of the input images, and its ability to generalize decreases when trained on small datasets with limited patient diversity, mainly due to overfitting and insufficient morphological variation, as noted in the original study.

### 4.2. Drug Development and Synthetic Biology

DeepSTARR [75] makes use of convolutional neural networks (CNNs) to directly predict enhancer activity from DNA sequences. There is a strong correlation of 0.86 between the prediction results and experimental data, which validates the model’s predictive power. This high level of accuracy supports the rational design of synthetic enhancers and facilitates the precise quantitative regulation of element activity. However, its reliance on STARR-seq experimental data restricts its generalizability. HyenaDNA [119] employs the Hyena operator as an alternative to traditional attention mechanisms, enabling efficient long-sequence modeling, which refers to learning dependencies across DNA sequences spanning tens to hundreds of thousands of base pairs. This capability is critical for identifying regulatory elements that may act at significant genomic distances. HyenaDNA [119] is designed to predict functional genomic annotations such as promoters, enhancers, and transcription factor binding sites from ultra-long nucleotide sequences (>100,000 bp). It achieves a Top-1 accuracy of 91%, meaning the correct functional label is ranked as the most probable among all possible classes. This performance was achieved on the Enformer benchmark dataset, which includes genome-wide annotations from the human reference genome. While HyenaDNA successfully captures distal regulatory signals beyond the scope of traditional attention-based models, it requires substantial computational resources due to the increased sequence length requirement and model complexity.

### 4.3. Precision Medicine and Health Management

DeepPT [17] is mainly built upon the Transformer architecture. It can predict genome-wide mRNA expression from histopathological images. When combined with the ENLIGHT model, it can predict the treatment response of cancer patients. ENLIGHT-DeepPT can predict patients’ responses to various treatments without any training on treatment data, achieving an overall advantage ratio of 2.28. Specifically, the response rate among patients predicted as responders increased by 39.5% compared to the baseline response rate observed in patients who did not receive ENLIGHT-matched treatments (i.e., those with an ENLIGHT matching score, EMS, below 0.54). The EMS quantifies the match between a patient and a given treatment based on the overall activation state of the genetic interaction (GI) partner genes of the drug targets, as inferred from gene expression data. It reflects the principle that tumors are more likely to respond to therapies that promote synthetic lethal (SL) interactions while avoiding synthetic rescue (SR) interactions. According to the original ENLIGHT study, patients with an EMS ≥ 0.54 are considered ENLIGHT-matched [17].

gemGAT [77] models the gene co-expression network topology based on Graph Attention Networks (GATs). In 47 tissues, 83% of its predictions outperform existing methods. Moreover, 91.49% of the tissue predictions have a correlation coefficient greater than 0.7, and it supports the prediction of rare tissue samples and has high clinical practicality. Nevertheless, it has a strong dependency on data quality, and its model stability needs improvement.

Gemini Ultra [24] enhances cancer classification accuracy through multimodal fusion (gene expression + pathological images), achieving a classification accuracy of 89% (compared to a single-mode baseline of 75%). Its cross-modal association enhances diagnostic reliability, but it relies on cloud computing, resulting in high deployment costs at the edge.

Basenji2 [120] is a convolutional neural network (CNN)-based model designed to predict the effects of local regulatory DNA sequences on gene expression. It operates on input sequences up to 131 kb in length but focuses primarily on proximal elements. When evaluated on the Geuvadis dataset [122], which includes genome-wide RNA-seq data from lymphoblastoid cell lines of 462 individuals in the 1000 Genomes Project, Basenji2 achieved a Pearson correlation coefficient of 0.85 between the predicted and observed gene expression levels. This indicates strong predictive power for nearby regulatory regions. However, the model does not account for distal regulatory elements, such as enhancers located hundreds of kilobases away, which limits its applicability in clinical contexts where long-range regulatory interactions are crucial.

## 5. Challenges in AI-Driven Prediction of Gene Expression

### 5.1. Data-Related Challenges: Bridging the Heterogeneity–Fusion Gap

The integration of multimodal data for gene expression prediction faces three major technical barriers: heterogeneity, noise, and high-dimensional features. In terms of modal heterogeneity, whole-slide histopathology images (WSIs) with millions of pixels require patch-based processing (e.g., 256 × 256 patches) followed by dimensionality reduction, creating cross-modal semantic gaps when aligned with genomic data (e.g., RNA-seq FPKM matrices containing 6 × 10^4^ genes) or text-based data (e.g., NLP-derived tumor-infiltrating lymphocyte (TIL) labels). Furthermore, spatiotemporal alignment remains critical for single-cell spatial transcriptomics (e.g., 10x Visium) to avoid cross-modal representation mismatches with imaging or textual data [123,124,125,126].

The high dimensionality of genomic data (e.g., RNA-seq matrices with ~60,000 genes) and imaging data (e.g., gigapixel WSIs) necessitates robust feature selection strategies and efficient dimensionality reduction techniques (e.g., PCA, autoencoders) to mitigate overfitting risks, alleviate computational bottlenecks, and retain biologically critical signals [124,125].

Data quality issues compound these challenges: staining artifacts in WSIs, amplification biases in RNA-seq, and subjective text annotations introduce multimodal noise. Cross-modal batch effects (e.g., sequencing platform variations with RSD > 30%) remain unresolved due to the lack of multimodal analogs to ComBat algorithms, necessitating joint modal alignment techniques for correction [127,128,129].

Current advancements in multimodal data integration primarily employ three computational strategies: Cross-modal alignment techniques are exemplified by spatial transcriptomic platforms like 10x Visium that enable single-cell resolution image–gene expression co-registration [130], alongside semantic embedding models that project histopathological features and omics data into unified latent spaces through contrastive learning [131]. Dynamic gated fusion architectures address modality dominance issues via learnable attention weights to balance morphological and genomic contributions [132], while dimensionality harmonization approaches leverage principal component analysis (PCA) [133] and variational autoencoders [134] to mitigate cross-modal heterogeneity in high-dimensional data. Future developments necessitate innovative frameworks integrating topological graph networks with Transformer architectures to resolve spatiotemporal dependencies in multi-scale tissue systems.

### 5.2. Model-Related Challenges: From Architectural Innovation to Interpretability

In biomedical research, multimodal fusion necessitates balancing cross-modal complementarity with biological specificity [135]. Early-fusion strategies (e.g., concatenating genomic and histopathological features) preserve raw data associations but risk overfitting in pan-cancer TCGA analyses (AUC declines by 5%) and may obscure independent biomarker effects [136]. Conversely, late-fusion approaches (e.g., integrating results from separate genomic and imaging analyses) enhance robustness but fail to capture spatial interactions between gene expression and CT imaging features in tumor microenvironments. Transformer-based cross-modal attention mechanisms dynamically link mutational profiles with imaging phenotypes yet require gated weighting to prevent genomic data from disproportionately dominating predictions [137,138].

Cross-modal representation learning often combines heterogeneous neural architectures, such as CNNs (for imaging) and BiLSTMs (for genomics) [139]. However, hybrid networks risk parameter explosion and overfitting [140]. Clinically, interpretability is paramount: techniques like Grad-CAM and SHAP values help quantify gene-level contributions, validating biological hypotheses [141]. Nevertheless, joint analyses of single-cell sequencing and whole-slide imaging generate terabyte-scale datasets, demanding model compression via knowledge distillation to meet real-time clinical diagnostic requirements.

### 5.3. Ethical and Legal Challenges

Even anonymized genomic data can re-identify individuals using ≥75 independent SNPs [142]. While the EU adopts differential privacy (expression error < 3%) to balance utility and security [143], GDPR-driven data sovereignty barriers hinder multinational collaboration. Accountability frameworks require clear agreements on cross-institutional data rights (hospitals, sequencing firms, publishers) and liability criteria for model errors based on validation rigor [144,145]. Ethical equity concerns arise from data biases, such as reduced melanoma detection sensitivity in African cohorts, partially addressable via adversarial debiasing. However, only 37% of countries have genomic data regulations, and existing laws (e.g., GINA) fail to address predictive misuse risks, underscoring the need for coordinated technical and regulatory innovation [146,147].

## 6. Prospects for the Future

As an emerging technology, AI holds immense potential in human pathology, particularly in gene expression prediction. Future directions should focus on the following:Innovation in Multimodal Fusion

Future research may benefit from the development of dynamic gated fusion architectures that can balance modality-specific characteristics—such as the high specificity of genomic data—with the complementary strengths of other modalities like imaging and textual features. A promising direction involves the integration of causal inference frameworks with graph neural networks (GNNs) to elucidate cross-modal biological mechanisms. For instance, such approaches could help unravel how non-coding variants influence histopathological phenotypes via protein interaction networks. Additionally, the incorporation of reinforcement learning (RL) may offer a flexible means of optimizing multimodal fusion strategies. By conceptualizing the fusion process as a sequential decision-making problem, RL can dynamically guide modality selection and integration based on task-specific rewards, such as improvements in predictive performance or model interpretability.

2.Interpretability Enhancement

To enhance interpretability in biomedical AI models, future efforts might explore the construction of biologically informed architectures grounded in existing knowledge graphs of gene regulatory pathways. Tools such as SHAP (Shapley Additive Explanations) could be employed to visualize the relative contribution of key features, including critical biomarkers like TP53 mutations. These predictions may be further validated through mechanistic experiments, such as CRISPR-based perturbations in patient-derived organoid models, thereby establishing a robust link between algorithmic inference and biological reality.

3.Data Governance and Ethical Frameworks

Ensuring the ethical deployment of AI in biomedicine will likely require advances in data governance and privacy-preserving collaboration mechanisms. One promising strategy is the deployment of heterogeneous federated learning systems, which facilitate multi-institutional cooperation without compromising patient privacy. Addressing data heterogeneity through domain adaptation techniques can further improve model robustness across sites. Moreover, establishing cross-racial fairness evaluation protocols—such as sensitivity compensation algorithms in melanoma detection for underrepresented populations—can inform the development of globally equitable AI systems. These efforts collectively contribute to a more inclusive and responsible landscape for genomic and medical AI research.

4.Clinical Translation Optimization

Lightweight hybrid models (e.g., Transformer-lite architectures) compatible with intraoperative real-time analysis (<500 ms inference time) could be engineered. Evidence-based mapping rules between AI predictions and clinical guidelines should be formalized, and multi-stakeholder governance boards should be established to delineate legal liability boundaries for algorithmic misdiagnoses involving high-risk biomarkers.

## 7. Conclusions

In summary, with the evolution of cross-modal pretrained models and generalist medical AI systems, gene expression prediction will progressively transition from a research tool to a pivotal clinical decision-support module. Through the synergistic advancement of technological innovation and ethical governance, AI-driven multimodal analytics holds promise for deciphering disease-specific molecular maps, ultimately realizing the “from pixels to genes” vision of closed-loop precision medicine.

## Figures and Tables

**Figure 1 biology-14-00651-f001:**
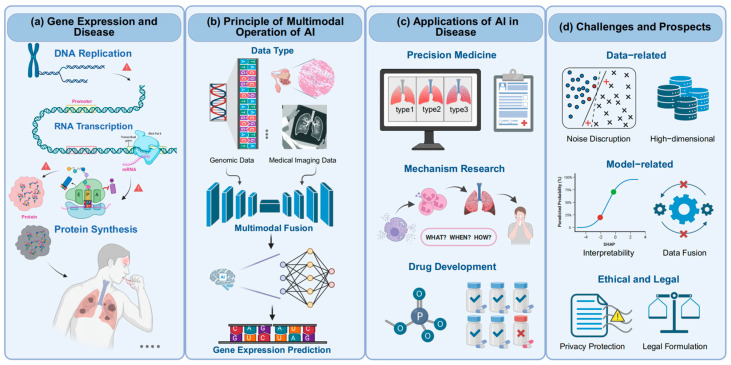
Schematic overview of AI-driven gene expression prediction in disease research. (**a**) Gene expression and disease. The gene expression process comprises DNA replication, RNA transcription, and protein synthesis; a disruption in any of these steps may lead to disease development. (**b**) Principle of multimodal operation of AI. Multimodal data, including genomic data, are fused through multimodal integration; then, machine learning and deep learning algorithms are used to predict gene expression. (**c**) Applications of AI in disease. Current gene expression prediction models have broad applications in precision medicine, the elucidation of disease mechanisms, drug development, and related fields. (**d**) Challenges and prospects. Key challenges in gene expression prediction encompass issues related to data quality, high-dimensional features, model interpretability, and difficulties in data alignment, as well as concerns surrounding privacy protection and ethical considerations. This figure was created with BioRender.com (accessed on 28 May 2025).

**Figure 2 biology-14-00651-f002:**
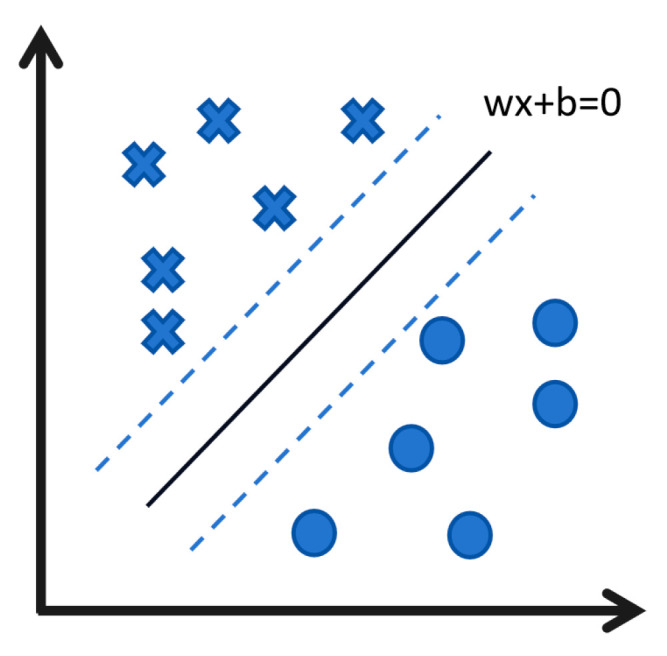
Support Vector Machine. The figure shows the distribution of one or more sample points in the feature space, as well as the optimal hyperplane learned by SVM. The ε-interval bands on both sides represent the error tolerance zone, which is equivalent to the safety margin. The dashed lines indicate the margins (i.e., the boundaries of the ε-interval) on either side of the hyperplane. Additionally, the vectors illustrated in the figure are the support vectors that define the hyperplane and determine its position.

**Figure 3 biology-14-00651-f003:**
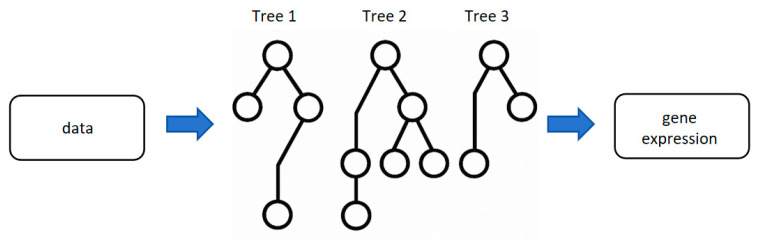
Random Forest. This figure shows the architecture of RF. Each tree is trained on a different subset of the input features, and the prediction results of each tree are combined by “voting” or “averaging” to improve the overall robustness. Each circular node represents a decision node based on a certain feature.

**Figure 4 biology-14-00651-f004:**
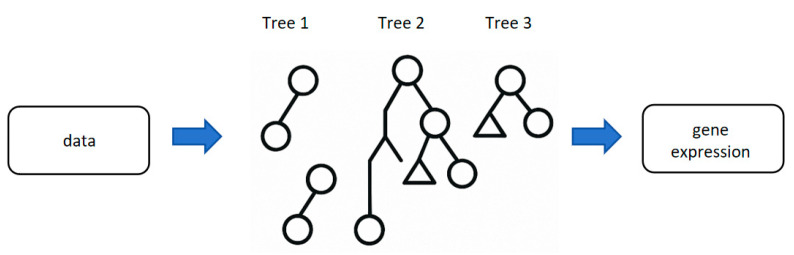
Gradient Boosting Machine. This figure shows the architecture of GBM. Each tree is not independently modeled but continues to fit based on the residual error of the previous tree, gradually improving the overall performance. Among them, the circular node represents the decision node, and the triangular node represents the residual adjustment node. The tree structure is split layer by layer from top to bottom, and finally, the prediction value increment is obtained.

**Figure 5 biology-14-00651-f005:**
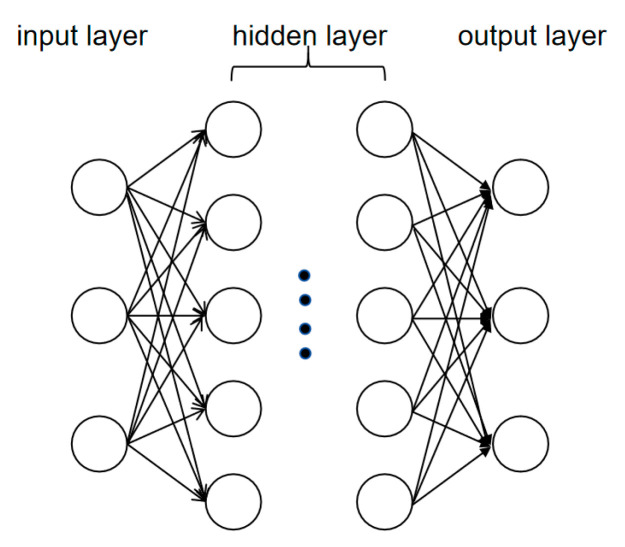
Convolutional neural network. This figure shows the architecture of a CNN. Each circular node in the figure is referred to as a neuron, which includes a weight matrix and an activation function. The model includes an input layer, a hidden layer, and an output layer. The hidden layer in the middle is also called a fully connected layer.

**Figure 6 biology-14-00651-f006:**
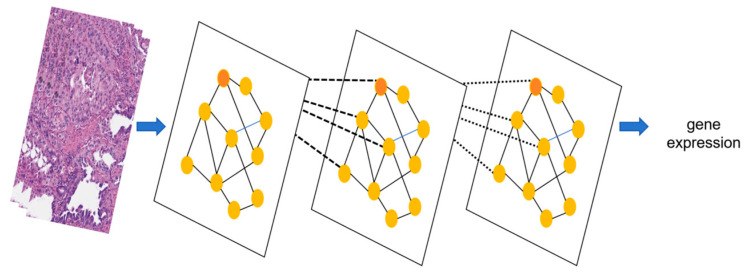
Graph neural network. This figure shows the architecture of a GNN. Each face is a graph where nodes represent genes or cells, and edges (lines) represent functional interactions between them, such as protein−protein binding or gene co−expression. This graph structure mirrors biological reality: genes and cells form networks with dynamic dependencies, such as signaling pathways in tumors. The GNN processes these interactions through iterative message−passing layers, enabling it to learn hierarchical patterns in gene expression data.

**Figure 7 biology-14-00651-f007:**
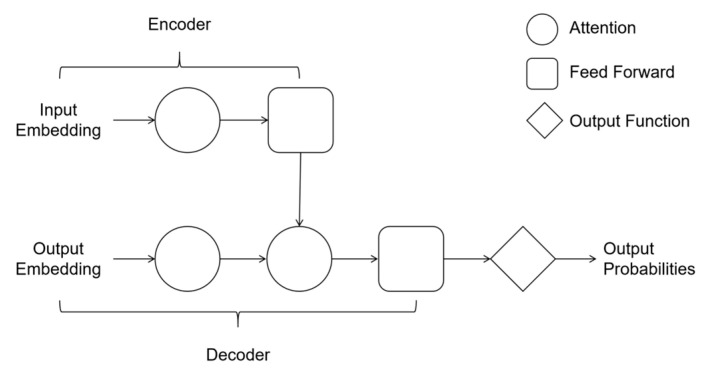
Transformer: This figure shows the architecture of Transformer. The input is omics data. A model is composed of multiple superimposed Transformer blocks. Each Transformer block consists of an encoder and a decoder to predict gene expression.

**Figure 8 biology-14-00651-f008:**
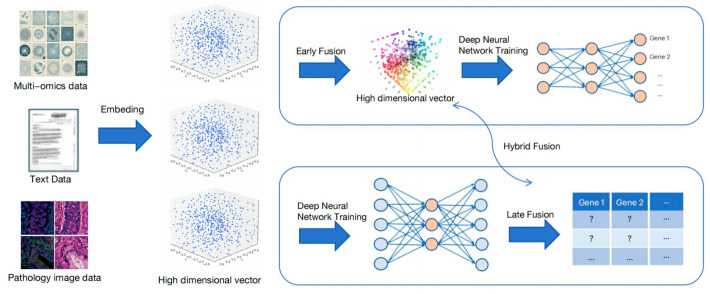
Multimodal model pipeline for predicting gene expression. Artificial intelligence integrates multimodal data to predict gene expression. Firstly, by embedding the model, multimodal data can be transformed into high-dimensional vectors. For early fusion, high-dimensional vectors are directly combined and fed into a neural network. For late-stage fusion, they are separately fused through neural networks. And mixed fusion is an alternating process between the two.

**Figure 9 biology-14-00651-f009:**
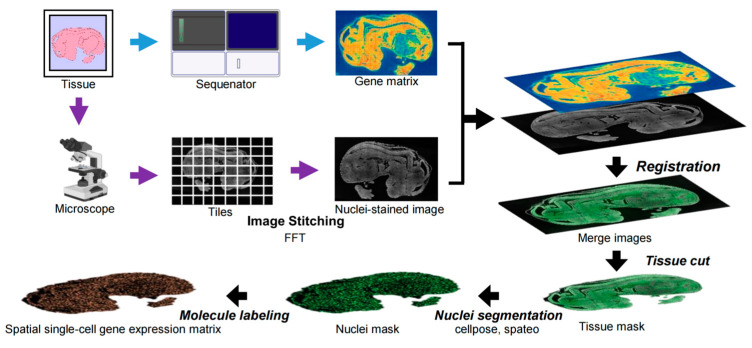
Schematic overview of the StereoCell workflow in [100]. Starting with tissue sections, a sequencing platform (e.g., Sequenator) is used to generate a spatial gene expression matrix. Simultaneously, high-resolution microscopic images of the tissue are captured and tiled, followed by image stitching using Fast Fourier Transform (FFT) to form a full mosaic image of the tissue. The gene expression matrix and the mosaic image are then co-registered to create a unified coordinate system. Using cell segmentation tools such as Cellpose [49] and Spateo [50], a nuclei mask and tissue mask are generated to identify individual cells and tissue regions (tissue cut). Finally, molecular labeling is performed, enabling the integrated extraction of spatial gene expression profiles and morphological features at single-cell resolution.

**Table 1 biology-14-00651-t001:** Unimodal data types and applications.

Data Type	Omics Data					Natural Language Data			Medical Imaging Data			
Biological Dimension	Molecular level: gene variation, epigenetics, RNA expression, protein interactions, metabolism					Clinical and knowledge-based: phenotypes, treatments, mechanisms, clinical notes			Structural and functional: organ morphology, tissue context, physiology, pathology			
Specific Type	Genomics	Epigenomics	Transcriptomics	Proteomics	Metabolomics	Scientific research literature	Medical records	Clinical reports	CT	MRI	PET	H&E
Data Resource	NCBI GeneBank	MethBank	GEO	Human Protein Atlas (HPA)	MetaboLights	PubMed	ClinicalTrials.gov	MIMIC-III	LIDC-IDRI	TCIA	Human Connectome Project (HCP)	TCGA
	UCSC Genome Browser	ENCODE (Encyclopedia of DNA Elements)	HCA (Human Cell Atlas)	PRIDE (Proteomics Identifications Database)	HMDB (Human Metabolome Database)	Google Scholar	/	MIMIC-IV	TCIA (The Cancer Imaging Archive)	/	/	CDSA (Cancer Slide Digital Archive)
	Ensembl Genome Browser	CNGB	GTEx	/	/	/	/	PhysioNet	/	/	/	GTEx Histology
	DDBJ	/	SpatialDB	/	/	/	/	/	/	/	/	/
	CNGB	/	TCGA	/	/	/	/	/	/	/	/	/

**Table 2 biology-14-00651-t002:** Artificial intelligence algorithms and their advantages.

Algorithms	Specific Technology	Advantages in Predicting Gene Expression	Current Applications in Clinical Pathology	Training Requirements for Medical Staff
Machine Learning Algorithms	SVM [56]	Proficient in handling high-dimensional small-sample data	Diagnosing cancer subtypes via gene expression analysis, aiding personalized treatment	Basic ML and data pre-processing, short-term online training suffices
	RF [57]	Strong ability to resist overfitting	Predicting cancer prognosis using clinical and genetic data	Feature selection and model evaluation knowledge, medium-level workshop training needed
	GBM [58]	Suitable for weak features in gene expression progressive learning	Predicting cancer treatment response	Gradient descent and parameter tuning, systematic courses and practice required
Deep Learning Algorithms	GNN [60]	Expert in relational modeling and analyzing biological networks	Predicting gene activity in rare tissues, understanding tumor microenvironments	Graph theory and GNN principles, specialized courses and hands-on training
	CNN [61]	Expert in extracting local sequence features	Analyzing histopathology for cancer subtype prediction and gene methylation for early detection	Image processing and CNN principles, medium-to-extensive case-based training
	Transformer [62]	Self-attention-based multimodal integration and long-range dependency modeling	Predicting gene expression across unseen cell types, integrating multi-omics data for diagnosis	Deep learning basics, Transformer architecture, long-term specialized training

**Table 3 biology-14-00651-t003:** Evaluation and clinical applications of AI models for gene expression prediction.

Model Name	Model Type	Application	Data Type	Advantages	Limitations
GET [20]	Transformer	Cross-cell-type gene prediction	Gene expression, chromatin accessibility, sequence data	Zero-shot prediction, multimodal integration	Data privacy and interpretability challenges
HGGEP [118]	GEM + HAM	Disease mechanism research, pathological image–gene correlation	Histological images	Visual association of tissue morphology and molecular mechanisms	Requires high-quality images, small-sample limitation
DeepSTARR [75]	CNN	Enhancer activity prediction	DNA sequences, STARR-seq data	Direct DNA sequence-to-enhancer activity prediction	Reliance on STARR-seq experimental data
HyenaDNA [119]	Hyena Operator-based Model	Long-range genomic sequence modeling, regulatory signal capture	DNA sequences, RNA secondary structure data	Single-nucleotide resolution, distal regulatory capture	High computational power requirement
DeepPT [17]	Transformer	Cancer treatment response prediction, transcriptomics simulation	H&E histopathology images, transcriptomics data	Non-invasive gene expression prediction from pathology images	Requires large, matched image–transcriptomics datasets
GemGAT [77]	Graph Attention Network (GAT)	Rare tissue gene expression prediction	Gene expression, spatial co-expression networks	Cross-tissue generalization, small-sample robustness	High data quality dependency, model stability issues
Gemini Ultra [24]	Multimodal Fusion Model	Multimodal cancer classification	Gene expression data, H&E pathology images	Cross-modal association for diagnostic reliability	High edge deployment cost, limited rare cancer subtype validation
Basenji2 [120]	CNN	Local regulatory sequence analysis	Genome sequences, chromatin states	Efficient promoter-proximal feature processing	Ignores distal regulatory elements

## Data Availability

Not applicable.

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
