# Peer review of "AI-Driven Transcriptome Prediction in Human Pathology: From Molecular Insights to Clinical Applications"

_biology, 2025, doi:10.3390/biology14060651_

Round 1

Reviewer 1 Report

Comments and Suggestions for Authors

In this article, Xiaoya et al. systematically reviewed AI application for predicting gene expression in human pathology. They found machine learning algorithms and deep learning models excel in extracting features from histopathology slides, genomic sequences. Before considering publishing in our journal, some revisions must be made.

Comments:

  1. The title of the manuscript need to change. Some model/tool do not tackle gene expression. For example: “SpliceAI[46] is based on deep convolutional networks (CNN) to analyze the impact of splicing variants on diseases”.  “Evo 2[48] capitalizes on ultra-large-scale genomic pre-training (9.3 trillion nucleo-238 tides) to analyze millions of sequences for the design of new molecules.”
  1. For a review article for model evaluation, a table of evaluation in main text is needed. Please put some of the results in Table S1 in to the main text.
  2. Please list the best used situation of each model/tool
  3. Please discuss how to ensure data privacy and ethical issues of AI models in clinical applications?
  4. How to tackle semantic differences between multiomics data, such as pathological images and gene expression
  5. Line 252, please bolder font of “3.3 Precision Medicine and Health Management”
  6. For publishing in such as reputable journal, only two table and one figure are unacceptable. Please make more tables or Figures, such as model classification, model difference.

Author Response

We sincerely apologize for not strictly following the MDPI template due to our oversight. However, we have made every effort to ensure that our responses are clear and well-organized. Please refer to the attached file for details.

Reviewer 2 Report

Comments and Suggestions for Authors

Chen et al have performed solid summary of currently available AI tools which can be applied for the prediction of the gene expression in human pathology. Authors have provided broad review of the currently available methods and have summarized the most recent AI models available up today. There are a couple of minor suggestions which authors could implement to improve the final version of the manuscript.

  • In section 2.1 (lines 91-99) Chen et al discuss different multi omics data. Authors may consider adding the proteomic level to the summary, due to the importance of studying the mechanism of the translation from mRNA to protein level
  • In section 2.1 (lines 109) authors should add the additional spatial omic section, where they could discuss mention spatial transcriptomic, spatial proteomics and how AI can be beneficial to connect H&E sections with these sophisticated spatial tools
  • In section 2.2, authors have discussed SVM, RF and GMB. It would strongly benefit if authors supplemented this section with references on the studies which use each of those mentioned machine learning approaches for human pathology applications.
  • Authors also can modify Table 2 adding references on pathology related studies using each of the mentioned technologies
  • Authors could consider adding the additional section to chapter 2 describing current advances in the application of the AI models in the clinical pathology and summarize how clinical pathologists can benefit from the usage of AI models and discuss which tools can be directly used and which tools would require extensive training for the medical personnel.
  • It will benefit to add to the manuscript the discussion about reinforcement learning and how it can be implemented in digital pathology and how it can complement the machine learning and deep learning tools (for instance in region-of-interest identification, adaptive scanning protocols, interactive diagnostic systems).
  • Supplementary material tables need to have references-there is a technical error caused disappearance of the references. Also, complement the S1 table with the model type column, i..e Model: DeepPT (CNN) etc

Author Response

(The authors gave the same response as above.)

Round 2

Reviewer 1 Report

Comments and Suggestions for Authors

The quality of manuscipt have hightly improved, now it can be accept

Author Response

Thank you for your feedback. We're glad to know the manuscript is now acceptable, and we truly appreciate your support and suggestions during the review process.